# The spatial and temporal differentiation characteristics of cultural heritage in the Yellow River Basin

Wei Li[1], Jianping Jiao[1]*, Jianwu Qi[2], Yujia Ma[1]

1 College of Geography and Environmental Science, Northwest Normal University, Lanzhou, China,
2 Institute of Urban Planning and Tourism Landscape Design, Northwest Normal University, Lanzhou, China

☯ These authors contributed equally to this work.
* jjpnwnu@163.com

**Data Availability Statement:** All relevant data are within the paper.

**Funding:** The Institute of Urban Planning and Tourism Landscape Design, Northwest Normal University provided material assistance for field research, and provided some financial support

## Abstract

Understanding the temporal and spatial distribution characteristics of the cultural heritage of the Yellow River Basin can effectively improve the scientific understanding of the historical changes, environmental evolution, and cultural and economic development of the Yellow River Basin and thus provide a scientific and reasonable decision-making basis for the protection and development of its cultural heritage. The research object of this paper are the national cultural relic protection units. These are examined using the GIS spatial analysis method to explore the spatial and temporal distribution characteristics and spatial structure of 2,102 national material cultural heritage sites in the Yellow River basin. The results show that the spatial distribution of cultural heritage has a significant spatial agglomeration effect. The whole basin is concentrated in stable high- and low-value areas, and the difference between the high- and low-value areas is clear. Some aspects of the spatial structure heterogeneity are strong, showing a low value dispersion distribution trend. In different periods, the distribution direction and scope of cultural heritage have low ranges of rotation, a clear direction, and a high degree of centripetal distribution. The spatial and temporal distribution of cultural heritage is the result of the combined action of natural geographical environment such as climate change, topography, river hydrology, and human environment such as administrative institutional changes, ideological evolution, and social and economic development.

## 1. Introduction

As the crystallization of the development of historical civilization, cultural heritage is an important symbol of a country or nation's historical and cultural achievements, and it is also an important cultural wealth [1]. Cultural heritage research is not only the inheritance of Chinese spiritual civilization, but also the protection of traditional culture [2]. The Yellow River is the most important birthplace of the Chinese nation and the cradle of Chinese culture [3]. Historical products of different times and spaces are recognized in a common time and space with a tendency toward national roots. Understanding and judging the spatial patterns and

during the data collection and analysis. The funders had no role in study design, data collection and analysis, decision to publish, or preparation of the manuscript. No authors receive salary support from funding units.

**Competing interests:** The authors have declared that no competing interests exist.

characteristics of cultural heritage in the Yellow River Basin has become an important issue for the development of spaces. Although the existing cultural heritage-related research in cultural heritage protection, cultural heritage assessment, cultural heritage management, and architectural heritage [4–8], is innumerable, most studies are limited to their respective domains in history, archeology, architecture, management, and social and cultural disciplines. The study of human geography in the field of cultural heritage is limited. Most of the existing research on cultural heritage focuses on urban areas, heritage distribution areas, cultural preservation units, cultural relics, and cross-regional cultural routes at the medium and micro scales, such as historical and cultural cities, towns, and villages; national, provincial, municipal, and county levels of cultural heritage; and important cultural relic distribution areas and cultural routes [9]. There is a lack of large-scale, macro- and regional distribution patterns and characteristics in cultural heritage research, and there are only a few studies on the cultural heritage of the Yellow River Basin at the watershed scale. As the product of time iteration and accumulation, cultural heritage has important spatial and temporal attributes and characteristics. It is particularly important to study the spatial and temporal distribution of the characteristics and patterns of cultural heritage. For example, domestic scholar Liao Lanqin studied the spatial and temporal characteristics and evolution of intangible cultural heritage in the Yangtze River Basin [10]. Li Fei analyzed the evolution of the spatial structure of linear cultural heritage from a theoretical framework and expounded upon the comprehensive impact on tourism [11]. Zhu Aiqin also studied the temporal and spatial evolution of cultural heritage in Hubei Province from the perspective of cultural heritage [12]. Bairushan explored the relationship between the spatial and temporal evolution characteristics of cultural heritage and the geographical environment and used Anhui Province as an example to illustrate the correlation between the distribution form of cultural heritage and topography, river systems, traffic foundations and historical accumulation [13].

However, in the existing literature, there lacks systematic and complete research on the spatial and temporal evolution, differentiation rules, driving factors, and influencing mechanisms of cultural heritage in the whole historical period of the Yellow River Basin. Furthermore, there still lacks research at the basin scale and this line of work needs further improvement and supplementation. As a representative integration of regional cultural heritage, different levels and immovable cultural protection units are an important part of regional material cultural heritage, reflecting the current situation of regional cultural protection and inheritance [14]. Eight provincial capital cities (Xining, Lanzhou, Yinchuan, Hohhot, Taiyuan, Xi'an, Zhengzhou, and Jinan) and nine provinces (Qinghai, Sichuan, Gansu, Ningxia, Inner Mongolia, Shanxi, Shaanxi, Henan, and Shandong) of the Yellow River Basin (Xining, Lanzhou, Yinchuan, Taiyuan, Xi'an, Zhengzhou, and Jinan), and 61 prefecture-level cities (prefectures, leagues) of national and provincial cultural relics protection units are the research objects of this paper. The spatial-temporal evolution and differentiation characteristics of cultural heritage in the historical period of the Yellow River Basin are studied in depth using the spatial analysis method. Investigating the differentiation patterns of cultural heritage at the watershed level allows deeper examination into the underlying issues of the analysis process. This study will serve as a reference for the high-quality development of material cultural heritage, intangible cultural heritage, and heritage tourism in the Yellow River Basin in the future [15–17].

## 2. Data sources and research methods

### 2.1. Data sources

Data on national key cultural relic protection units in the Yellow River Basin were derived from the list of the first eight batches of national key cultural relic protection units published

by the official website of the State Cultural Relics Bureau and the websites of the nine provincial and district cultural relic bureaus in the Yellow River Basin (as of October 2019). According to the declaration lists of the different national key cultural relic protection units, the 2,102 national protection units in the Yellow River Basin are divided into six categories: 1) ancient ruins, 2) grotto temples and stone carvings, 3) ancient tombs, 4) ancient buildings, 5) modern important historical sites and representative buildings, and 6) other types. The coordinate data of these key cultural relics were obtained from Google Earth and cross-checked using the Baidu map coordinate picking system. The vector data of administrative divisions, boundaries, rivers, and lakes in the Yellow River Basin were downloaded from the geospatial data cloud, and the administrative boundary data of the Yellow River Basin were cut and calibrated using the ArcGIS10.2 map processing tool. The resulting 601 counties of 84 prefecture-level cities in the Yellow River Basin were identified as the research units.

## 2.2. Research method

**2.2.1. Kernel density estimation.**    The kernel density estimation method can intuitively reflect the spatial distribution characteristics of point elements in the region [18]. This paper uses the kernel density estimation method to express the spatial distribution density and local characteristics of material cultural heritage in the Yellow River Basin. The formula is as follows:

$$\hat{\lambda}_h(s) = \sum_{i=1}^{n} \frac{3}{\pi h^2} \left[ 1 - \frac{(s - s_i)^2}{h^2} \right]^2 \tag{1}$$

where S is the position of the object to be estimated, S is the position of the first estimation object in the circle with S as the center and h as the radius and the value of h will affect the smoothness of the spatial distribution of the kernel density of the estimated object.

**2.2.2. Exploratory spatial data analysis (ESDA).**    Exploratory spatial data analysis (ESDA) is a spatial analysis method that reveals the spatial correlation and heterogeneity between research objects through the description and visualization of spatial distribution patterns and is based on mathematical statistics and graphic expression. The methodology comprises global spatial autocorrelation analysis and local spatial autocorrelation analysis [19].

Global autocorrelation is a description of the spatial characteristics of the whole region, reflecting the similarity of the observed values of spatially adjacent region units. Moran's I index is commonly used and ranges from -1 to 1. If Moran's I index is greater than 0 or less than 1, then there is positive spatial correlation. If 'Moran's I index is less than 0 or more than -1, then there is negative spatial correlation. If Moran's I index is equal to 0, then there is no spatial correlation. In this paper, Moran's I is used to represent the spatial aggregation degree of cultural heritage.

The global Moran's I calculation formula is as follows:

$$I = \frac{\sum_{i=1}^{n} \sum_{i \neq j}^{n} (x_i - \overline{x})(x_j - \overline{x})}{S^2 \sum_{i=1}^{n} \sum_{i \neq j}^{n} W_{ij}} \tag{2}$$

In (2), $S^2 = \frac{1}{n} \sum_{i=1}^{n} (x_i - \overline{x})^2$, $\overline{x} = \frac{1}{n} x_i$ is the attribute value of position i and position j. This paper expresses the number of cultural heritage sites in different historical periods in the Yellow River Basin. The spatial weight matrix is generally used to test whether there is spatial

autocorrelation between regions. The formula is as follows:

$$Z = \frac{I - E(I)}{\sqrt{VAR(I)}} \tag{3}$$

Local spatial autocorrelation can measure the spatial distribution and interaction of cultural heritage in different geographical units. Commonly used methods are Moran scatterplots and Moran's I statistics. The local Moran's I index is as follows:

$$I_i = \frac{(x_i - \overline{x})}{S^2} \sum_{i \neq j}^{n} W_{ij} \left( x_j - \overline{x} \right) \tag{4}$$

In Formula 4, n,Xi,Xj, and Wij are the same as in Formula 2. The greater the absolute value of Ii is, the higher the local spatial correlation is. The test formula of Ii is as follows:

$$Z = \frac{I - E(I_i)}{\sqrt{VAR(I_i)}} \tag{5}$$

**2.2.3. Standard deviation ellipse.** Standard deviation ellipses can be used to aggregate and express spatial features such as central, discrete, and directional trends of geographical elements. They can be used to reveal the spatial distribution of cultural heritage in the Yellow River Basin and indicate changes in its central position and movement trends [20]. The formula is defined as follows:

$$C = \frac{1}{n} \left( \begin{matrix} \sum_{i=1}^{n} \overline{x}_i^2 & \sum_{i=1}^{n} \overline{x}_i \overline{y}_i \\ \sum_{i=1}^{n} \overline{x}_i \overline{y}_i & \sum_{i=1}^{n} \overline{y}_i^2 \end{matrix} \right), \begin{cases} \left( x_i - \overline{x'} \right) \\ \left( y_i - \overline{y'} \right) \end{cases} \tag{6}$$

In Formula 6, Xi and Yi are the coordinates of element i,$(\overline{X'}\ \overline{Y'})$ which is the mean center of the element, and n is the total number of elements.

# 3. Temporal and spatial characteristics of cultural heritage by period

According to the historical development process of Chinese civilization and the differences in administrative division boundaries in different historical periods, Chinese history is divided into seven periods: 1) the pre-Qin period, 2) the Qin and Han dynasties, 2) the Wei, Jin, Southern, and Northern Dynasties, 3) the Sui, Tang, and Five Dynasties, 4) the Song and Yuan dynasties, 5) the Ming and Qing dynasties and 6) modern times. These periods are used for studying the spatial and temporal distribution characteristics of national key cultural relic protection units in the Yellow River Basin across time. Dynasties that run through different historical periods in the historical evolution of Chinese civilization have been divided in recent years [21].

Chinese civilization from the pre-Qin period to the founding of various cultural relics spans up to 1.7 million years of history. First, the pre-Qin period was the embryonic stage of the development of Chinese civilization. Chinese history moved from cave dwellings to settled farming, then from the Stone Age to the Iron Age, and from ignorance and barbarism to civilization [22]. During this period, the national cultural protection units in the Yellow River Basin were mainly ancient tombs, followed by ancient buildings, which laid the foundation of Chinese civilization. The Qin and Han dynasties and the Wei, Jin, Southern and Northern

Dynasties were the stages marked by prosperity and transformation of Chinese civilization. At this stage, the internal political situation was turbulent, and marked by warfare. China's administrative system developed over the subsequent Qin, Eastern Han, Western Han, Three Kingdoms, Two Jin, Southern, Northern, and other dynasties. The cultural relics were mainly ancient sites, ancient tombs, grotto temples, and stone carvings, followed by ancient buildings. The Sui, Tang, and Five Dynasties and the Song and Yuan Dynasties were the shaping stages of Chinese civilization. During this period, Chinese history experienced the prosperity and development of the Sui and Tang Dynasties, the split of the Five Dynasties and Ten Kingdoms, and then the unified prosperity and development during the Song and Yuan Dynasties. Cultural relics are mainly grotto temples, stone carvings, and ancient buildings, followed by ancient sites, and cultural relics are developing toward diversification [23]. From the Ming and Qing Dynasties to modern times is the regeneration and prosperity stage of Chinese civilization. The cultural relics in this stage were mainly ancient buildings, important historical sites, and representative buildings in modern times, including the very famous classical garden architecture relics in the Ming and Qing Dynasties and some military sites, red sites, and industrial relics in modern times [24]. Spatial distribution points of different types of cultural heritage in each period were determined by spatial analysis based on watershed boundaries (Fig 1).

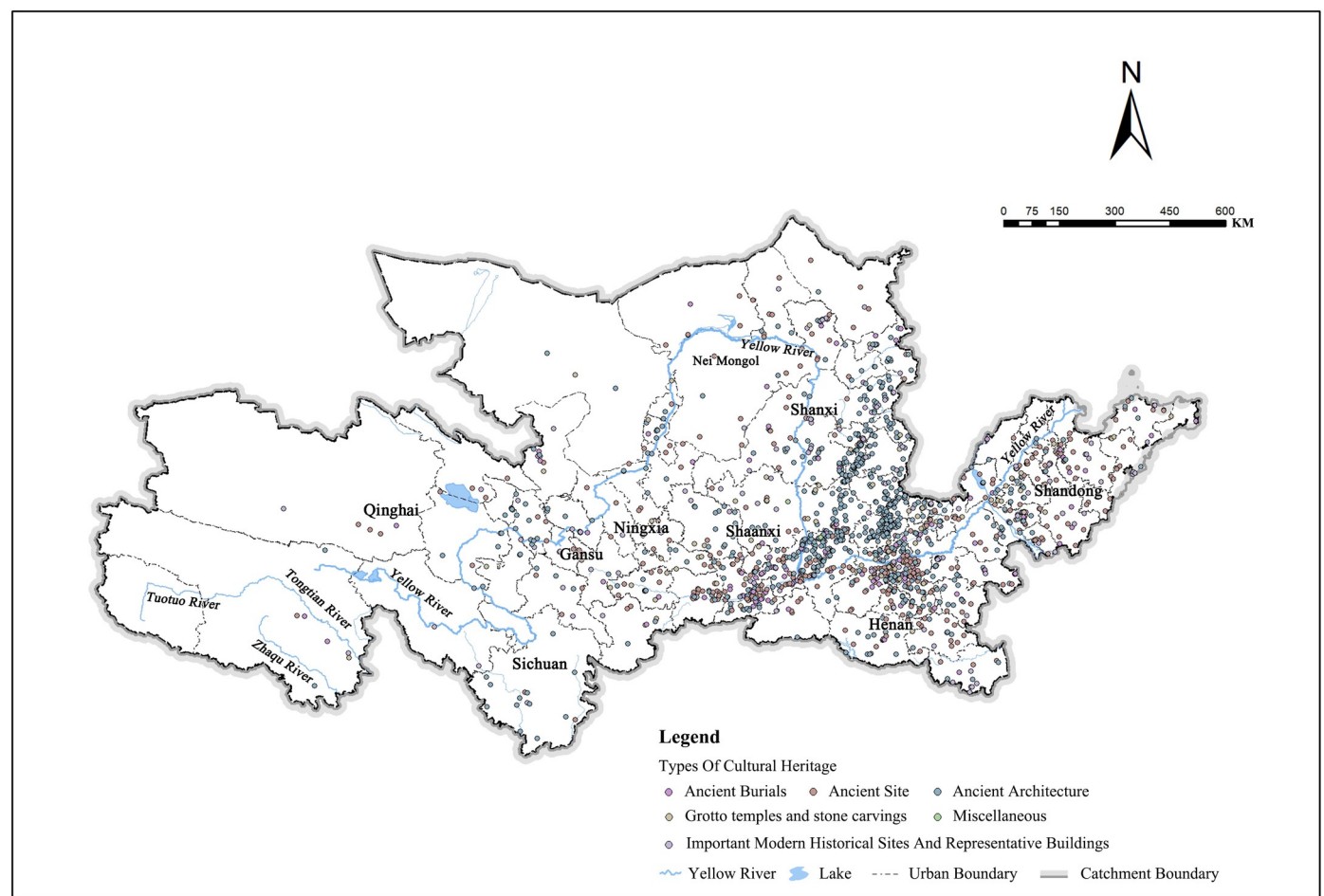

**Fig 1. Spatial distribution points of various types of cultural heritages in the Yellow River Basin.**

### 3.1. Spatial and temporal distribution characteristics of each historical period

Within the Yellow River Basin, the number and types of cultural heritage in each historical period are different. Therefore, according to the number of cultural heritage in each period, the proportion of the total heritage and the historical cognition of each period to summarize and The distribution characteristics and spatial evolution direction of cultural heritage in each historical period were judged (Table 1).

### 3.2. The overall spatial and temporal distribution characteristics of the Yellow River Basin

The spatial distribution of cultural heritage sites in the Yellow River Basin was visualized using kernel density analysis (Fig 2). Cultural heritage sites are primarily distributed across the Song and Yuan Dynasties and the Ming and Qing Dynasties, and those from the Ming and Qing Dynasties are the most prominent. The cultural heritage sites of the Yellow River Basin are different across the historical periods, but they are relatively concentrated in their respective distribution areas of the era, and their overall trend rebounds in the historical period [25–31]. Along the spatial dimension, the cultural heritage sites of the Yellow River Basin are characterized by large agglomeration and small dispersion. From pre-Qin to modern times, cultural heritage sites show a trend of transferring from the west, northwest, east, south, and southeast. Mainly distributed in the Guanzhong Basin, the North Henan Plain, and the Northwest Shandong area as the core and spread outward. Among them, the number of cultural heritages in the Wei, Jin, Southern and Northern Dynasties decreased sharply, and there was a trend of transferring from the Guanzhong Basin to Taiyuan and Jinan. Among them, the number of cultural heritages in the Wei, Jin, Southern and Northern Dynasties decreased sharply, mainly in grotto temples and stone carvings, and there was a trend of transferring from the Guanzhong Basin to Taiyuan and Jinan. Since then, in various stages of historical development from the Sui, Tang and Five Dynasties to the Ming and Qing Dynasties, the overall number of cultural heritage shows a trend of fluctuation and rebound. During the Ming and Qing Dynasties, the cultural heritage was the largest, mainly ancient buildings, concentrated in Shanxi. In the modern and modern historical period, the number of cultural heritage is relatively small due to the short historical process and the period of great change and turmoil throughout the country.

Overall, cultural heritage sites of the Yellow River Basin across the historical periods are primarily distributed in the middle and lower reaches of the Yellow River, especially in the middle and lower reaches of the plains, where agricultural civilization was relatively developed, as well as the Guanzhong Basin and other places. The population grew steadily and has a clear urban orientation. Generally, the closer to modern times the era is, the greater the number of cultural heritage sites there are and the greater awareness of respecting and protecting cultural heritage people have [32–38].

## 4. Evolution of spatial patterns of cultural heritage sites in the Yellow River Basin

### 4.1. Spatial structure characteristics in different periods

**4.1.1. Spatial agglomeration characteristics are clear, and cultural heritage space shows a significant positive correlation.** The global Moran's I value of the distribution quantity index in different historical periods of the Yellow River Basin was calculated by using exploratory spatial statistical analysis (ESDA) and Geo Da analysis software (Table 2) to explore the

**Table 1. Distribution characteristics of cultural heritage in different periods.**

| Historic Stage | Amount of heritage (P) | Share Of The Total (%) | Historical Cognition | Distribution characteristics |
|---|---|---|---|---|
| Pre-Qin Period | 361 | 17.17 | Start-up development stage. It spanned many historical forms such as the Paleolithic Age to the Spring and Autumn Period and the Warring States Period. Due to the long period of time, the low level of productivity and more wars, the number of preserved cultural heritage is relatively small, and most of them are ancient ruins. | 1. In Shaanxi Guanzhong (Baoji, Xi'an)-Shanxi (Yuncheng) area, it is distributed in a belt shape. |
| | | | | 2. Centralized in the northern region of Henan, Zhengzhou as the center of the cluster distribution. |
| | | | | 3. Form small agglomeration centers in Jinan, Zibo, Jining and other places in Shandong. |
| Qin-Han Period | 180 | 8.56 | Stage of prosperity. It was the period of the initial development of our feudal society and the first appearance of a unified situation. The influence of the separatist forces of the Qin and Han regimes led to a sharp increase in the number of ancient sites used for strategic defense and military preparations. | 1. The Guanzhong area radiating from the capital of the Western Han Dynasty (Xi'an) as the center. |
| | | | | 2. It is concentrated in Luoyang, the capital of the Eastern Han Dynasty, and the northern part of Henan with Zhengzhou as the center, showing a polar nucleus-diffusion distribution. |
| | | | | 3. Centered in Jining and Tai'an, Shandong, and concentrated in a small area. |
| Wei, Jin, Southern, and Northern Dynasties | 119 | 5.66 | Prosperity and evolution stage. This period was the period of the division of the feudal state and the great integration of nationalities in my country. Buddhism prevailed, rulers used religion to build temples, carved cave statues, and the number of cave temples and stone carvings increased rapidly. | 1. With Luoyang as the center, it extends to Jincheng in Shanxi and Anyang in Henan. |
| | | | | 2. Taking Tai'an, Shandong as the center, extending to Jining and other directions. |
| | | | | 3. A smaller agglomeration center in Taiyuan, Shanxi. |
| | | | | 4. In Tianshui and Pingliang in Gansu Province, Guyuan in Ningxia, and Guanzhong in Shaanxi, there are scattered distribution patterns. |
| Sui, Tang, and Five Dynasties | 189 | 8.99 | Definite form and transformation stage. This period was the stage when my country's feudal society reached its peak and then split. Inscriptions and grottoes and pagodas form style-specific grotto features and artistic styles. | 1. With Xi'an as the core, it spreads around. |
| | | | | 2. Centered on Luoyang and Zhengzhou in central Henan, it spreads to the surrounding area and Jincheng, Shanxi. |
| | | | | 3. It spreads around Jinan and Tai ' an in Shandong Province. |
| | | | | 4. Smaller agglomeration centers are formed in Anyang in northern Henan and Taiyuan in Shanxi. |
| Song-Yuan Period | 513 | 24.41 | Definite form and transformation stage. This period was the period when the feudal country was divided, and the national regimes were established side by side to gradually realize the great unification. The economic center of gravity has been moved to the south, the exchanges between China and foreign countries are frequent, and the number of ancient buildings has increased sharply. | 1. Most of them are concentrated in Shanxi Province, with Yuncheng, Jincheng, Taiyuan, Changzhi and other places as the core, and are distributed contiguously. |
| | | | | 2. Larger agglomeration centers are also formed in central Henan, northern Henan and northwestern Shandong. |
| | | | | 3. It is scattered in a small amount in Longzhong of Gansu Province, Ningxia and Inner Mongolia. |
| Ming-Qing Dynasty | 710 | 33.78 | Prosperity regeneration stage. This period was the stage of the gradual decline of my country's feudal system and the further consolidation of the unified multi-ethnic state. Clan temples, ancestral halls, ancient residential buildings, and frontier defense projects are on the rise, and the number of ancient buildings accounts for the absolute majority. | 1. It is highly concentrated in the Shanxi region, extending from the Taihang Mountains to the Guanzhong Basin, and is densely distributed in a band. |
| | | | | 2. In Jincheng and Changzhi of Shanxi Province also form large dense centers, showing contiguous distribution. |
| | | | | 3. With Luoyang as the center, it spreads to the surrounding areas, and also forms smaller agglomeration centers in Jining and Jinan, Shandong. |
| | | | | 4. The number of cultural heritage in Qinghai, Ningxia, Gansu, Sichuan, Inner Mongolia and other places has increased significantly. |

(*Continued*)

**Table 1.** (Continued)

| Historic Stage | Amount of heritage (P) | Share Of The Total (%) | Historical Cognition | Distribution characteristics |
|---|---|---|---|---|
| Modern Period | 179 | 8.52 | This period is an evolutionary stage with a short development process, newer cultural heritage and less quantity. It mainly takes modern practical activities as the main body, has a clear purpose and urban orientation, and takes the important historical sites and representative buildings in modern times as the absolute main body. | 1. Taking modern revolutionary activities as the main axis, it is scattered in various revolutionary practice places in the Yellow River Basin, and has a clear purpose and urban orientation. |
| | | | | 2. Small agglomeration cores are formed in Longzhong, Jinbei, Jinzhong, Jinnan, Yuzhong, Henan and southeastern Shandong. |

spatial evolution and internal relations of cultural heritage sites in the Yellow River Basin across different historical periods. Overall, the Moran's I value of the Yellow River Basin was greater than zero, and the test statistic of the 'Moran's I value was greater than the critical Z value (1.96) at the 0.05 confidence level. This indicates that the cultural heritage in different historical periods of the Yellow River Basin has obvious and positive spatial autocorrelation characteristics, i.e., the distribution has obvious spatial dependence and obvious spatial agglomeration characteristics (Fig 3) [39]. The spatial agglomeration of cultural heritage in different historical periods of the Yellow River Basin is characterized by three types: 1) the strong agglomeration form in the Song and Yuan Dynasties, 2) the Ming and Qing Dynasties, and 3) the Pre-Qin Dynasty (Moran's I: 0.2419–0.4346). The general agglomeration forms in the Sui, Tang, and Five Dynasties (Moran's I: 0.1922), and the weak agglomeration forms in the Qin and Han Dynasties; the Wei, Jin, the Southern, and Northern Dynasties; and in modern times (Moran's I: 0.0594–0.0894) [40].

**4.1.2. The local agglomeration is dominated by one-way agglomeration in a stable positive correlation high-value area, and the spatial agglomeration effect is strong.** The study found that the pre-Qin period, Song and Yuan Dynasties, and Ming and Qing Dynasties show the characteristics of significant high-value agglomeration, i.e., high-value and high-value (H-H) agglomeration (Table 3). A stable positive correlation is the main type of agglomeration across these three historical stages. The area of high-value cultural heritage sites in the basin has evolutionary characteristics similar to those of its adjacent counties. The spatial agglomeration effect is strong, and the hierarchical diffusion structure is clear. These are also the three periods with the most obvious spatial agglomeration characteristics [41–43].

The H-H type county units in the pre-Qin period accounted for 45.74% of the local autocorrelation in the historical period, forming a strong H-H type spatial agglomeration area. H-H county units are also concentrated in the middle and lower reaches of the basin, in which Sanmenxia City and Zhengzhou City form a large-scale high-value agglomeration area. A medium-sized agglomeration area is also formed between Jining City and Tai'an City in Shandong Province;. A small-scale high-value agglomeration area is formed at the intersection of Zibo City, Dongying City, and Weifang City in Shandong Province.

The H-H type county units in the Song and Yuan Dynasties and the Ming and Qing Dynasties accounted for 55.91% and 52.78%, respectively, of the local autocorrelation in this historical period, accounting for the absolute main body and forming a strong H-H type spatial agglomeration area. Most of the high-value agglomeration areas in these two periods are concentrated in Shanxi Province, and they form large-scale contiguous agglomeration areas in Yuncheng, Jincheng, Jinzhong, and Changzhi. In addition, in the Song and Yuan dynasties, high-value agglomeration areas of sub-level scale appeared in some areas where Zhengzhou and Xuchang met.

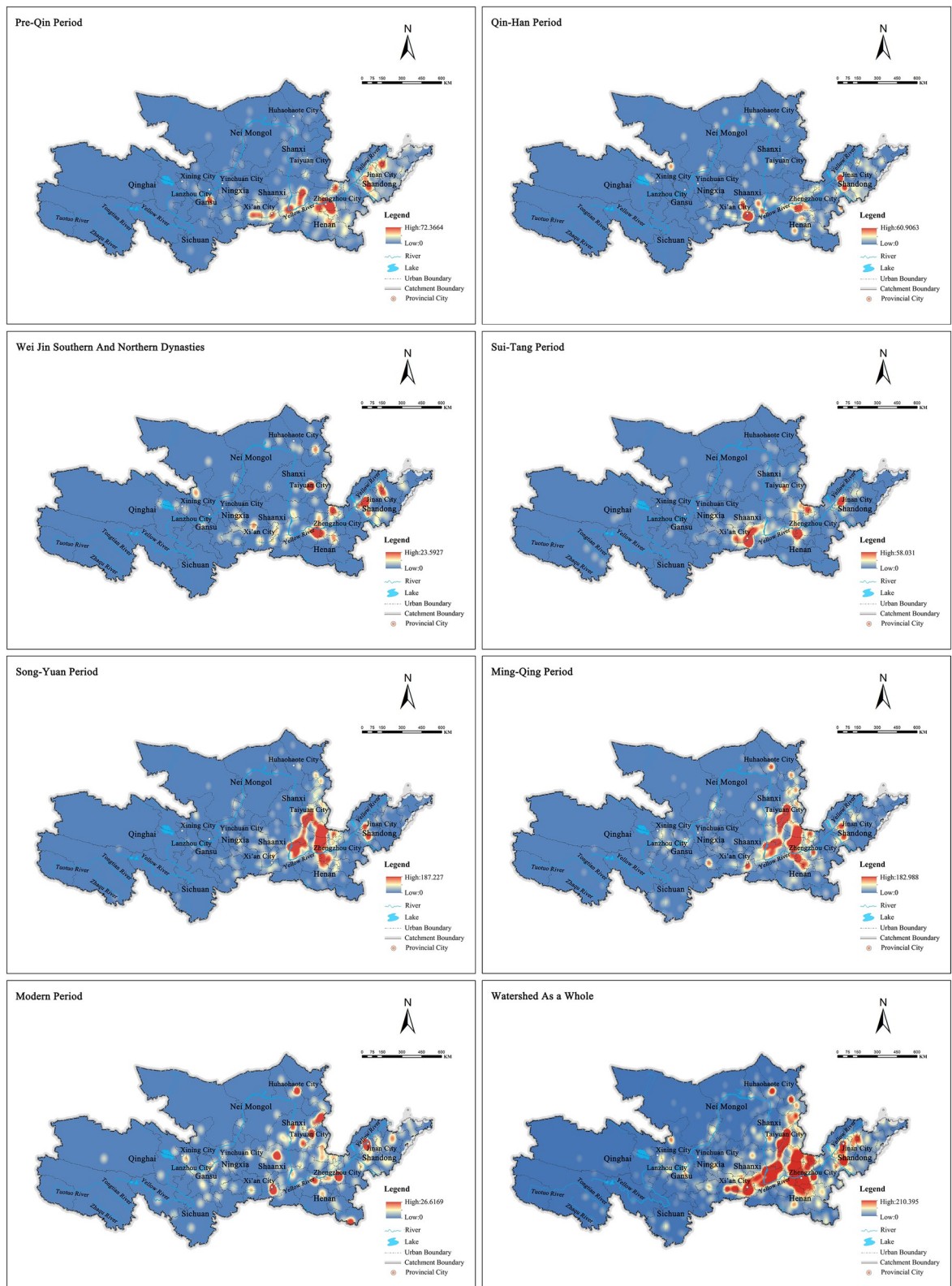

**Fig 2. Distribution map of cultural heritage nuclear density in different historical periods.**

**Table 2. Moran's I index and test of cultural heritage in different historical periods.**

| Historic Stage | Moran's I | E[I] | Mean | sd | P value | Z value |
|---|---|---|---|---|---|---|
| Pre-Qin Period | 0.2419 | -0.0014 | -0.0010 | 0.0221 | 0.0010 | 10.1872 |
| Qin-Han Period | 0.0894 | -0.0014 | -0.0016 | 0.0236 | 0.0030 | 3.8565 |
| Wei, Jin, Southern, and Northern Dynasties | 0.0594 | -0.0014 | -0.0019 | 0.0220 | 0.0140 | 2.7871 |
| Sui, Tang, and Five Dynasties | 0.1922 | -0.0014 | -0.0024 | 0.0226 | 0.0010 | 8.5919 |
| Song-Yuan Period | 0.4346 | -0.0014 | -0.0015 | 0.0220 | 0.0010 | 19.823 |
| Ming-Qing Dynasty | 0.3925 | -0.0014 | -0.0016 | 0.0233 | 0.0010 | 16.8786 |
| Modern Period | 0.0770 | -0.0014 | -0.0010 | 0.0235 | 0.0090 | 3.3206 |
| Overall Yellow River Basin | 0.3659 | -0.0014 | -0.0017 | 0.0228 | 0.0010 | 16.1186 |

**4.1.3. The heterogeneity of some spatial structures is strong, showing a low value scattered distribution.** In the historical stages represented by the Qin and Han Dynasties; the Wei, Jin, Southern and Northern Dynasties; the Sui, Tang, and Five Dynasties; and modern times, the spatial structure tends to be dispersed, showing no obvious characteristics of indigenous agglomeration. The spatial distribution type is mainly low-value and high-value agglomeration (L-H). Many counties show a negative correlation, indicating that the spatial heterogeneity of cultural heritage sites is strong in these three historical stages and showing a discrete distribution trend. Agglomerations of L-H type are mainly scattered in the local counties of various provinces in the Yellow River Basin. There is little correlation with the county units that have a high number of cultural heritage developments in their surrounding area (and thus becomes low-value, isolated areas). This evolution trend is highly consistent with the number and distribution characteristics of cultural heritage sites in the Yellow River Basin across the three historical periods. This trend also verifies that cultural heritage sites in the turbulent period are scattered, and the spatial agglomeration effect is weak [44–46].

**4.1.4. Overall, in a stable high- and low-value area, the differences in two-way agglomeration and high- and low-value are distinct.** Through cluster analysis of cultural heritage sites of the Yellow River Basin in all periods, the basin as a whole shows significant high and low value agglomeration characteristics, high and high value (H-H) versus low and low value (L-L) agglomeration. The H-H agglomeration area accounts for 41.38% of the spatial agglomeration type and is mainly distributed in the middle and lower reaches of the Yellow River. The cultural heritage sites represented by ancient buildings are concentrated in the areas connected between Yuncheng, Jincheng, Changzhi, and middle Shanxi. The spatial agglomeration effect is clear and shows significant positive correlation characteristics of agglomeration, while the surrounding counties and cities show similar evolutionary characteristics. The L-L type county unit accounted for 37.24%, which was second only to the H-H type, thereby indicating that a large part of the low value cultural heritage agglomeration in the Yellow River Basin is surrounded by other low value areas. Thus, cultural heritage is similar to the surrounding low value agglomeration. Overall, the positive correlation between H-H high-value agglomeration and L-L low-value agglomeration in the Yellow River Basin accounted for 78.62% of the cultural heritage agglomeration types in the whole basin. This shows that the overall spatial agglomeration form of the basin shows significant high and low spatial differences. There are similar evolution characteristics in the space of high-value areas, low-value areas, and other adjacent areas. The relationship between counties shows a significant hierarchical diffusion structure. The H-H type agglomeration area is mainly concentrated in the middle and lower reaches of the Yellow River. Its high-value agglomeration pattern confirms the evolution characteristics of population, production, and life in each historical stage of the Yellow River Basin. The L-L-type agglomeration area

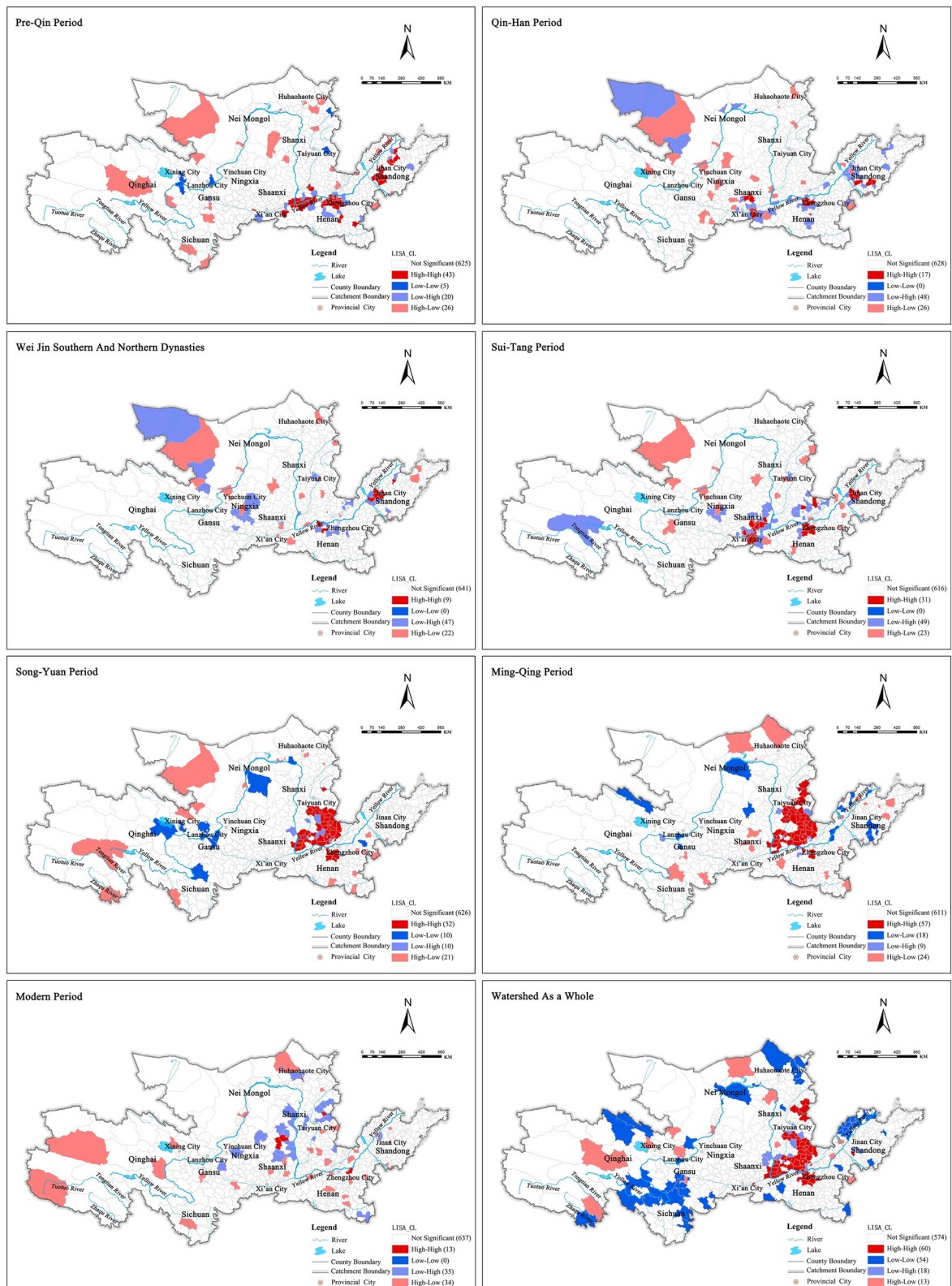

**Fig 3. Spatial Lisa distribution of cultural heritage in different historical periods.**

**Table 3. Spatial agglomeration types and proportions in different historical periods.**

| Agglomeration Type | Pre-Qin Period | Qin-Han Period | Wei, Jin, Southern, And Northern Dynasties | Sui, Tang, and Five Dynasties | Song-Yuan Period | Ming-Qing Dynasty | Modern Period | Basin as a whole |
|---|---|---|---|---|---|---|---|---|
| H-H | 45.74% | 18.68% | 11.54% | 30.09% | 55.91% | 52.78% | 15.85% | 41.38% |
| L-L | 5.32% | 0 | 0 | 0 | 10.75% | 16.67% | 0 | 37.24% |
| L-H | 21.28% | 52.75% | 60.26% | 47.57% | 10.75% | 8.33% | 42.68% | 12.41% |
| H-L | 27.66% | 28.57% | 28.21% | 22.33% | 22.58% | 22.22% | 41.46% | 8.97% |

is mainly scattered in marginal areas with a low development level, small populations, and few traces of social production and life throughout history [47–50].

## 4.2. Distribution and spatial evolution trend of cultural heritage centers in different periods

**4.2.1. The distribution direction of each period varies, and the center of gravity is roughly similar.** The standard deviation ellipse tool was used to calculate the distribution direction of cultural heritage in each historical period and obtain the spatial distribution center of gravity of each period. The results show that in the historical evolution of the seven regions, the overall trend is from southwest to northeast and from west to east (Fig 4). The distribution center of gravity in each historical period is roughly similar, and the center of gravity trajectory is continuously distributed. The focus of cultural heritage in the pre-Qin period was in Yangcheng County, Jincheng City, Shanxi Province (112.62˚E, 35.41˚N), covering approximately 67.31% of the cultural heritage sites of that period. During the Qin and Han Dynasties, the center of cultural heritage shifted northeastward to the area of Qinshui County (112.29˚ E, 35.59˚ N), north of Jincheng City, Shanxi Province. The offset distance was 50.35 km, which affected approximately 72.77% of the cultural heritage during this period. During the Wei, Jin, Southern, and Northern Dynasties, the focus of cultural heritage shifted northwestward to Hongdong County, Linfen City, Shanxi Province (112.05˚E, 36.20˚N), by 103.73 km, affecting approximately 68.91% of the cultural heritage sites of that period. During the Sui, Tang, and Five Dynasties, the focus of cultural heritage shifted southwestward to the junction of Hancheng City in Weinan City of Shaanxi Province and Hejin City in Yuncheng City of Shanxi Province (110.88˚E, 35.58˚N), with an offset distance of 147.47 km, which affected approximately 70.37% of cultural heritage in this period. During the Song and Yuan Dynasties, the center of cultural heritage moved northeastward again to Yaodu District of Linfen City, Shanxi Province, with an offset distance of 123.08 km, which affected approximately 69.60% of the cultural heritage of the period. During the Ming and Qing Dynasties, the focus of cultural heritage shifted to Xiangning County, Linfen City, Shanxi Province, which contained approximately 67.46% of the cultural heritage during this period. In modern times, the center of cultural heritage moved northeastward to Yaodu District, Linfen City, Shanxi Province, with an offset distance of 53.03 km, which affected approximately 67.46% of the cultural heritage sites of the period (Table 4). In general, the center of the distribution of cultural heritage sites in the Yellow River Basin is similar in each period, and the trajectory of the center of gravity changes near Linfen City, Shanxi Province [51].

**4.2.2. The turning angle of cultural heritage in each period is low, directionality is clear, and centripetal distribution is high.** From the change in ellipse direction in each historical period, the rotation angle is relatively low, moving in the range of 85.80˚ to 93.27˚. The rotation angle first increases and then weakens before suddenly and significantly decreasing. The long axis of the ellipse changes between 1142.52 km and 1065.86 km, while the short axis of the ellipse changes between 505.21 km and 600.93km. From the Sui, Tang, and Five Dynasties

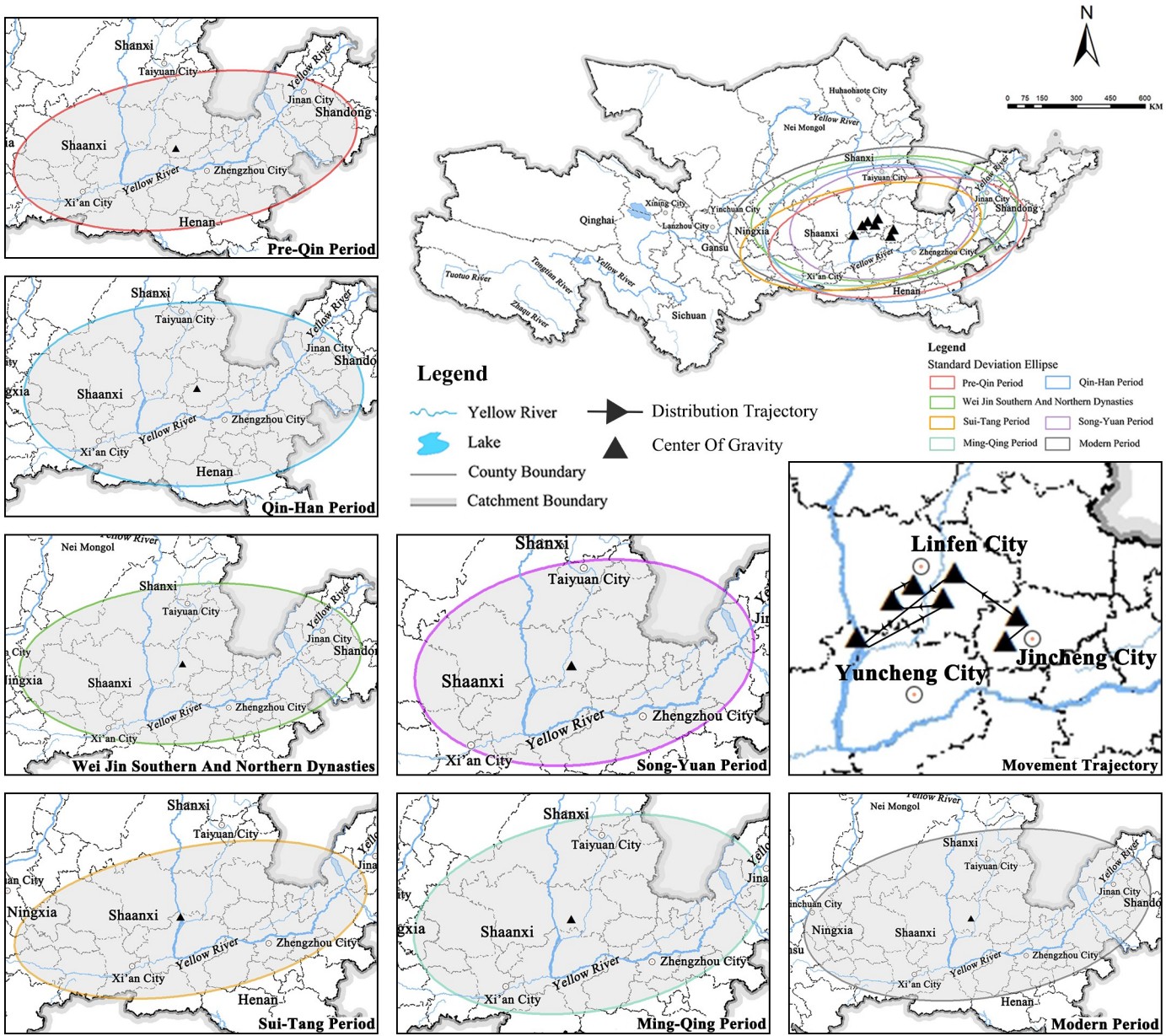

**Fig 4. The focus and direction of cultural heritage distribution in different periods.**

periods to the modern period, the amplitude of the rotation angle in the elliptical direction remains stable within a range from 86.34˚ to 88.21˚. The semi-major axis of the ellipse fluctuates between 812.24 km and 1280.27 km. The short half axis of the ellipse floats between 486.18 km and 607.48 km. Compared with the modern period, the difference in long-short half-axis values between the Song and Yuan dynasties and the Ming and Qing dynasties is shorter, indicating that the elliptical distribution flat rate is small. This also implies that the distribution direction of cultural heritage is unclear and the degree of dispersion is low. In modern times, the gap between the long axis and the short axis is large, and the flat rate in the elliptical direction is also large, indicating that the direction of cultural heritage distribution in this historical stage is clear and that the degree of discrete distribution is high.

**Table 4. Variation in standard deviation ellipse parameters in different historical periods.**

| Historic Stage | Areal Coordinates | Directional angle | Long Half Axis | Short Half Axis | Moving Direction | Range | Offset Distance |
|---|---|---|---|---|---|---|---|
| Pre-Qin Period | 112.62°E,35.41°N | 87.96° | 1142.52Km | 505.21Km | Southwest-Northeast | 67.31% | - |
| QinHan Period | 112.29°E,35.59°N | 93.27° | 1104.97Km | 600.93Km | West-East | 72.22% | 50.35Km |
| Wei, Jin, Southern, and Northern Dynasties | 112.05°E,36.20°N | 90.95° | 1167.61Km | 538.65Km | West-East | 68.91% | 103.73Km |
| Sui, Tang, and Five Dynasties | 110.88°E,35.58°N | 85.80° | 1065.86Km | 437.01Km | Southwest-Northeast | 70.37% | 147.47Km |
| Song-Yuan Period | 111.96°E,36.02°N | 88.21° | 812.24Km | 486.18Km | Southwest-Northeast | 69.60% | 123.08Km |
| Ming-Qing Dynasty | 111.56°E,36.09°N | 86.34° | 987.69Km | 537.88Km | Southwest-Northeast | 67.46% | 79.68Km |
| Modern Period | 112.62°E,35.41°N | 87.96° | 1280.27Km | 607.48Km | Southwest-Northeast | 64.80% | 53.03Km |

Overall, the distribution direction of cultural heritage sites in each historical period shows a distribution pattern of diffusion-agglomeration-diffusion-agglomeration-agglomeration-diffusion-rediffusion, with a small angle and a total offset distance of 557.34 km. The overall distribution range is relatively stable, and the elliptical variation range shows a moving trajectory from diffusion to agglomeration and then to diffusion around Linfen City and Jincheng City in Shanxi Province [52].

## 5. Discussion

In this paper, the spatial and temporal characteristics of cultural heritage in the Yellow River Basin were analyzed in detail through spatial analysis. The spatial representation of cultural heritage in each historical period was found, and the focus and evolution direction of cultural heritage distribution in each period were summarized. In the process of research and analysis, it is found that the temporal and spatial evolution of cultural heritage is the result of the combined action mechanism of natural and human factors. Among them, human factors have a dominant force on the distribution characteristics of cultural heritage in the Yellow River Basin, while natural factors have a certain supporting and restricting effect on the distribution of cultural heritage in the Yellow River Basin.

### 5.1. Cultural heritage distribution trend is closely related to terrain climate coupling relationship

In historical evolution, terrain landforms and climate environments have a close coupling relationship with the cultural heritage distribution. The distribution core of cultural heritage is mainly located in Guanzhong Basin, North Henan Plain, along the Taihang Mountains and the western and northern alluvial plains. There is less cultural heritage in the Tibetan Plateau, Inner Mongolia Plateau and the Loess Plateau in terms of terrain landform. Areas with relatively low terrain such as valleys, plains and basins located in the main stream of the Yellow River and the alluvial areas of tributaries are not only the core of population production and intensive urban development in history, but also the density core of the distribution of many cultural heritages. At the same time, the climate plays an important role in historical processes, restricts and supports human production activities, and is also an important factor that leads to the prosperity and recession of cultural heritage in each time. In the historical period of warm period, the traces of population activities and production are obvious, and the inheritance and development of culture are also in increasing trend. In the historical evolution stage of the cold period, the traces of population activities are greatly reduced, not only in agricultural production, but also in the stagnation or reduction stage of cultural transmission.

## 5.2. River hydrology is an important factor in the agglomeration of cultural heritage

Water resources have always been an important source of human life and production, and the life and production of human populations have revolved around rivers since the beginning of human life. The distribution density and evolution of cultural heritage in the Yellow River Basin are directly related to the river system. In various historical stages of our country, many cities and capitals are built along the river, especially in the middle and lower reaches of the Yellow River where the terrain is relatively flat. Many cities and populations in the historical stage are derived from this, and a rich and colorful cultural heritage has been nurtured. Since the pre-Qin period, the distribution of population and life and production have also revolved around the main stream of the Yellow River and its tributaries at all levels, cultural heritage shows the distribution pattern along the river strip. In history, the Yellow River Basin has always been the birthplace of many cultures and an important memory place for the derivation of Chinese culture. The excellent traditional culture represented by Yangshao Culture, Shanxi Merchant Culture, Longshan Culture, Qilu Culture, Dadiwan Culture, etc. has been cultivated within the river basin, with rich historical and cultural heritage. Moreover, the distribution of cultural heritage also changes with the changes of rivers. In history, the multiple diversions of the main stream of the Yellow River have profoundly affected the social and economic environment at that time, making the distribution of cultural heritage also follow the changes of ancient river courses. The overall cultural heritage shows a trend of "distribution along the river".

## 5.3. Administrative system and social economy are the dominant forces in heritage distribution

The changes in the national administrative system and socio-economic development in history have had a profound impact on the changes in the quantity of cultural heritage, and the rise and fall of dynasties has witnessed the degree of cultural prosperity in various historical periods. The number of cultural heritages in the Yellow River Basin has experienced a sharp decrease from the Qin and Han Dynasties and the Wei, Jin, Southern and Northern Dynasties, to the fluctuation and recovery of the Sui, Tang and Five Dynasties, to a sharp increase in the Song and Yuan Dynasties and the Ming and Qing Dynasties. The reason is that the establishment and consolidation of the authoritarian system in the Qin and Han dynasties, the long-term division of the country during the Wei, Jin, Southern and Northern Dynasties, and the division of war and chaos, greatly reduced the number of cultural heritage. The Sui, Tang and Five Dynasties were the heyday of my country's feudal society. The frequent exchanges between China and foreign countries and the vigorous development of social economy and various cultures have led to an increasing trend in the number of cultural heritage. The feudal state in the Song and Yuan Dynasties went from being divided to being united gradually, during this period, the economic center of gravity was moved to the south, the commodity economy had new development, the exchanges between China and foreign countries were frequent, overseas trade was developed, and science and technology, literature and art, and private trade were unprecedentedly prosperous, resulting in a substantial increase in the number of cultural heritage. In the Ming and Qing dynasties, the unified multi-ethnic state was further consolidated, the feudal economy developed unprecedentedly, the commodity economy was unprecedentedly prosperous, capitalism emerged and developed slowly, and the number of cultural heritage was further increased.

In a word, in each historical period of the Yellow River Basin, when the national strength was prosperous, the social and economic development was active, and the ideology and culture

were enlightened, the number of cultural heritages had obvious quantitative advantages. How-
ever, in the historical period of frequent wars, rapid regime change, social and economic
downturn, and conservative ideology and culture, the number of cultural heritage is relatively
small, and to a certain extent, the remnants of other cultures are inhibited.

## 6. Conclusion

Based on the data of 8 batches of national security units published by the state, the distribution
characteristics and evolution process of national security units in the historical period of the
Yellow River Basin were studied by using ArcGIS spatial analysis, kernel density analysis, spa-
tial local autocorrelation (ESDA), and standard deviation ellipse. The results of the spatial and
temporal analysis of cultural heritage sites in the Yellow River Basin have the following
implications.

1. The cultural heritage in the Yellow River Basin spans a long history. Cultural heritage sites
   were mostly concentrated in the Song and Yuan Dynasties and the period after the Ming
   and Qing Dynasties, which together account for 66.70% of the total, and of which the Ming
   and Qing Dynasties were the most prominent. The number and types of cultural heritage
   sites in the Yellow River Basin are different across the distinct historical periods, but there
   is a relatively concentrated distribution within each period. In general, cultural heritage
   shows a fluctuating upward trend at each historical stage.

2. In spatial dimension, the cultural heritage of the Yellow River Basin shows a distribution
   trend of large agglomeration and small dispersion. The spatial distribution type is mainly
   condensed distribution, and large condensed centers are formed around Luoyang, Xi'an,
   Taiyuan, Jincheng, Kaifeng, Jinan and other cities in the middle and lower reaches of the
   Yellow River. In general, the cultural heritages of the Yellow River Basin are primarily dis-
   tributed in the middle and lower reaches of the Yellow River, especially in its plains, where
   agricultural civilization was relatively developed, as well as in the Guanzhong Basin and
   other places. The population evolution process is continuous and showed clear river-side
   distribution patterns.

3. The overall spatial agglomeration characteristics of cultural heritage spatial structure in the
   Yellow River Basin areclear, and the cultural heritage space is positively correlated. There
   are strong agglomeration patterns in the Yellow River Basin as a whole, and especially in
   the Song and Yuan dynasties, the Ming and Qing dynasties, and the pre-Qin period. There
   was a general agglomeration pattern in the Sui, Tang, and Five Dynasties, and a weak
   agglomeration pattern in the Qin and Han dynasties, the Wei, Jin, Southern, and Northern
   dynasties, and modern times. The local one-way agglomeration is dominated by stable posi-
   tive correlation high-value areas, and the spatial agglomeration effect is strong. Some spatial
   structure heterogeneity is strong, showing a low-value dispersion distribution. Overall, the
   basin has stable high- and low-value areas that exhibit two-way agglomeration, and high-
   and low-value differences are distinct.

4. The center of cultural heritage in the Yellow River Basin from the pre-Qin period to the
   modern period is in Shanxi. In each period, the range of cultural heritage rotation is low,
   the direction isclear, and the centripetal distribution is high. In the evolution process of the
   seven historical periods, the overall evolution trend is from southwest to northeast and
   from west to east. The distribution center of gravity in each historical period is roughly sim-
   ilar, and the center of gravity trajectory is continuously distributed. Overall, the distribution
   direction of cultural heritage in each historical period shows a diffusion-agglomeration-

diffusion-agglomeration-agglomeration-diffusion-rediffusion distribution pattern, with a total offset distance of 557.34 km. The change of the overall distribution range is relatively stable, and the change range of the center of gravity is around Linfen City and Jincheng City in Shanxi Province, in which there was a small range of diffusion to agglomeration and then to diffusion.

## Acknowledgments

On the occasion of the completion of my thesis, I would like to thank Dr. Jian Liu for his warm care and careful guidance. In the process of writing my thesis, whether it is in the process of writing the article for help in the research framework, data analysis and writing revision, or in the research methods of the paper and the finalization of the paper, I have received Dr. Jian Liu's careful and meticulous teaching and selfless help, and I would like to express my sincere thanks and deep gratitude.

At the same time, I also thank my classmates Haozhou Fang, Chunyue Zhang and Kai Zhang, who provided great help in my paper data collection, data processing and result analysis, laying the foundation for the smooth completion and quality improvement of the article. In addition, I would like to express my sincere gratitude to the School of Urban Planning and Tourism Landscape Design of Northwest Normal University for its support in the collection of papers and other aspects.

Finally, I would like to express my heartfelt thanks to the experts who took the time out of their busy schedules to review this article and provide valuable comments!

## Author Contributions

**Conceptualization:** Wei Li, Jianping Jiao.

**Data curation:** Jianwu Qi.

**Formal analysis:** Yujia Ma.

**Funding acquisition:** Wei Li, Jianping Jiao.

**Investigation:** Yujia Ma.

**Methodology:** Wei Li, Jianping Jiao.

**Project administration:** Wei Li, Yujia Ma.

**Resources:** Yujia Ma.

**Software:** Jianwu Qi.

**Supervision:** Jianping Jiao.

**Validation:** Jianping Jiao.

**Visualization:** Jianwu Qi.

**Writing – original draft:** Wei Li.

**Writing – review & editing:** Wei Li, Jianping Jiao.

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
