## [Decision Letter · Decision Letter 0]

30 Dec 2021

PONE-D-21-38789The spatial and temporal differentiation characteristics of cultural heritage in the Yellow River BasinPLOS ONE

Dear Dr. Jiao,

Thank you for submitting your manuscript to PLOS ONE. After careful consideration, we feel that it has merit but does not fully meet PLOS ONE’s publication criteria as it currently stands. Therefore, we invite you to submit a revised version of the manuscript that addresses the points raised during the review process.

ACADEMIC EDITOR:The study has value but the manuscript has many problems as suggested by the reviewers. The authors should respond to the comments of the reviewers one by one and revise the manuscript accordingly. The revised manuscript might be sent to the reviewers for further reviewing.

We look forward to receiving your revised manuscript.

Kind regards,

Jian Liu

Academic Editor

PLOS ONE

Journal Requirements:

4. PLOS requires an ORCID iD for the corresponding author in Editorial Manager on papers submitted after December 6th, 2016. Please ensure that you have an ORCID iD and that it is validated in Editorial Manager. To do this, go to ‘Update my Information’ (in the upper left-hand corner of the main menu), and click on the Fetch/Validate link next to the ORCID field. This will take you to the ORCID site and allow you to create a new iD or authenticate a pre-existing iD in Editorial Manager. Please see the following video for instructions on linking an ORCID iD to your Editorial Manager account: https://www.youtube.com/watch?v=_xcclfuvtxQ.

5. We note that Figures 1-4 in your submission contain [map/satellite] images which may be copyrighted. All PLOS content is published under the Creative Commons Attribution License (CC BY 4.0), which means that the manuscript, images, and Supporting Information files will be freely available online, and any third party is permitted to access, download, copy, distribute, and use these materials in any way, even commercially, with proper attribution. For these reasons, we cannot publish previously copyrighted maps or satellite images created using proprietary data, such as Google software (Google Maps, Street View, and Earth). For more information, see our copyright guidelines: http://journals.plos.org/plosone/s/licenses-and-copyright.

a) You may seek permission from the original copyright holder of Figure 1-4 to publish the content specifically under the CC BY 4.0 license.  

Natural Earth (public domain): http://www.naturalearthdata.com/.

6. We note you have included a table to which you do not refer in the text of your manuscript. Please ensure that you refer to Table 5 in your text; if accepted, production will need this reference to link the reader to the Table.

7. Please upload a copy of Supporting Information Figures 1-4  and Supporting information Tables 1-5 which you refer to in your text on page 24.

Reviewers' comments:

Reviewer's Responses to Questions

**Comments to the Author**

1. Is the manuscript technically sound, and do the data support the conclusions?

Reviewer #1: Yes

Reviewer #2: Partly

2. Has the statistical analysis been performed appropriately and rigorously? 

Reviewer #1: Yes

Reviewer #2: I Don't Know

3. Have the authors made all data underlying the findings in their manuscript fully available?

Reviewer #1: Yes

Reviewer #2: Yes

4. Is the manuscript presented in an intelligible fashion and written in standard English?

Reviewer #1: Yes

Reviewer #2: No

5. Review Comments to the Author

Reviewer #1: This research aims for an interesting topic to explore the spatial and temporal characteristics of cultural heritage in the Yellow River Basin.

Overall, the paper is well written, but there are still some major concerns related to this work:

Although the paper pays attention only to the changes and distribution characteristics of cultural heritage, it is still necessary to explain and discuss some data in detail: why did the number of cultural heritages increase sharply from the Song and Yuan dynasties?

The selection basis of different periods needs to be further elaborated: The Ming and Qing Dynasties were more than 100 years longer than the Song and Yuan Dynasties, and this period has been much longer than the modern period.

The paper uses a lot of space to introduce the composition of cultural heritage in different dynasties, but this part is not reflected in the abstract: It is needed to add a diagram that shows different parts and also their linkages and relationships.

Although the paper has proposed to conduct in-depth research and analysis on the influencing factors in future research, it is still suggested to add some discussion on the results in the paper, there is too little discussion about the results.

Accordingly, I will suggest to have a strong improvement.

Reviewer #2: The spatial and temporal differentiation characteristics of cultural heritage

in the Yellow River Basin

The article carries out a in-depth spatial analysis of the spatial distribution pattern of 2102 cultural assets in the Yellow River Basin, across seven historical macroperiods and divided in six basic site categories, ranging from palaeolithic to modern times. The declared purpose (in the abstract) is to "serve as a reference for the high quality development and protection of cultural heritage in the Yellow River Basin" (a rather vague goal, in my view). Eventually, the paper defines through procedures of spatial analysis a sequence of fluctuating agglomeration and secondary dispersion.

The main weak points of this articles are the following:

1. The script is too long and often redundant. For example, paragraphs 3.1 to 3.7 can be easily

summarized in one or more tables, limiting the text to the most crucial comments. It is also very demanding in terms of illustrations (Figures 2, 3 and 4, in particular, would need a lot of space, probably pages, to be fully readable).

2. Overall, its english is quite poor. Many expressions are probably literal, or almost literal translations of chinese constructions which, once translated, remain obscure. For example - just to give an example, consider this sentence (first page of the text)

"Cultural heritage research is an important part of building cultural confidence, building regional cultural security patterns and Chinese cultural identity systems, and supporting high-quality regional development in the new era."

One guesses that "regional cultural security patterns" refers to practices of protection of cultural heritage by the state, but this is far from clear. And what exactly is "the new era"?

And consider this sentence (from Data sources):

"Data on national key cultural relic protection units in the Yellow River Basin were derived from the list of the first eight batches of national key cultural relic protection units published by the official website (http://www.ncha.gov.cn/) of the State Cultural Relics Bureau (hereinafter referred to as the State Protection) and the websites of the nine provincial and district cultural relic bureaus in the Yellow River Basin..."

Unreadable. Possibly, the "national key cultural relic protection units" are departments or organizations of the Ministry of Cultural Heritage, but this could be expressed much better.

And what are the "red sites" at the end of paragraph 3????

In short, my impression is that - as far as points 1 and 2 are concerned - the text should be shortened by 50% and the english reviewed if not re-written by a mother language speaker.

3. The Authors should attenuate some nationalistic overtones of their text. In particular, although the fact the the Yellow River Basin was the cradle of the Chinese civilization is universally accepted, sentences of the same content are unnecessarily repeated in different parts of the article (see in particular paragraph 3).

4. Similarly, I find disturbing some very idealistic assumptions - like the statement that "Cultural heritage (is) the crystallization of the development of human civilization" or the concept of a "Chinese spiritual civilization" (see the first page). Such expressions, I fear, have nothing to do with science.

5. The Authors have an evolutionary, firmly stadial concept of history. Instead of using archaeological patterns for reconstructing history, the give history for granted (the discussion of every historical period stats with a summary interpretation of an evolutionary step) and proceed to discuss the locational variations of sites.

Moreover, one of their periods is not credible: what is the historical sense of a pre-Qin stage, which goes from Paleolithic (1.700.000 mya, see paragraph 3.1) to the Qin dynasty (late 3rd century BC)? And sites which are linked to Revolution events, can be really analyzed in the same conceptual and analytical frame of prehistoric or protohistoric sites?

6. Even though the spatial analysis seems to be accurate and interesting, the evolution of settlement patterns in the study area is represented in terms of simple dimensional point patterns in a simple void space. No geomorphology, no remote sensing, no paleofluvial variations, no climatic changes, no changing patterns of soil exploitation, no demography. What about changing production systems? How did a type of economical practices of a phase affected those of the following period? What happened to wood coverings and soil formations? This is what, in my view, makes the article very idealistic and far from a materialist historical effort.

6. PLOS authors have the option to publish the peer review history of their article (what does this mean?). If published, this will include your full peer review and any attached files.

Reviewer #1: No

Reviewer #2: No

---

## [Author Response · Author response to Decision Letter 0]

16 Mar 2022

Academic editor:

1. Please ensure that your manuscript meets PLOS ONE's style requirements, including those for file naming. The PLOS ONE style templates can be found at https://journals.plos.org/plosone/s/file?id=wjVg/PLOSOne_formatting_sample_main_body.pdf and https://journals.plos.org/plosone/s/file?id=ba62/PLOSOne_formatting_sample_title_authors_affiliations.pdf.

Answer:

First of all, I am very grateful to the reviewers for their valuable comments, I will make the following reply：

In this revision, I have corrected the full text according to the format requirements of your journal. 

Answer:

First of all, I am very grateful to the reviewers for their valuable comments, I will make the following reply：

When the first draft of this paper was submitted, funding information and financial disclosure had been submitted, but the National Social Science Fund information was not queried in the options. I will try to submit this information again . 

Answer:

First of all, I am very grateful to the reviewers for their valuable comments, I will make the following reply：

Please keep the information statement submitted in the first draft and do not need to change it again。

4. PLOS requires an ORCID iD for the corresponding author in Editorial Manager on papers submitted after December 6th, 2016. Please ensure that you have an ORCID iD and that it is validated in Editorial Manager. To do this, go to ‘Update my Information’ (in the upper left-hand corner of the main menu), and click on the Fetch/Validate link next to the ORCID field. This will take you to the ORCID site and allow you to create a new iD or authenticate a pre-existing iD in Editorial Manager. Please see the following video for instructions on linking an ORCID iD to your Editorial Manager account: https://www.youtube.com/watch?v=_xcclfuvtxQ.

Answer:

First of all, I am very grateful to the reviewers for their valuable comments, I will make the following reply：

A new ORCID iD has been created in this modification, and the information update has been completed. 

5. We note that Figures 1-4 in your submission contain [map/satellite] images which may be copyrighted. All PLOS content is published under the Creative Commons Attribution License (CC BY 4.0), which means that the manuscript, images, and Supporting Information files will be freely available online, and any third party is permitted to access, download, copy, distribute, and use these materials in any way, even commercially, with proper attribution. For these reasons, we cannot publish previously copyrighted maps or satellite images created using proprietary data, such as Google software (Google Maps, Street View, and Earth). For more information, see our copyright guidelines: http://journals.plos.org/plosone/s/licenses-and-copyright.

a)You may seek permission from the original copyright holder of Figure 1-4 to publish the content specifically under the CC BY 4.0 license.

Answer:

First of all, I am very grateful to the reviewers for their valuable comments, I will make the following reply：

Figures 1-4 are completely drawn by the author. In Figures 1-4 mentioned in the submission of the first draft of this paper, the boundary range of the Yellow River Basin is obtained for free download from the "Geospatial Data Cloud" website, processed by myself in Arc Gis, and then superimposed the required cultural heritage spatial data Points, and then drawn through spatial analysis, do not involve copyright issues, you can use it with confidence. 

6.We note you have included a table to which you do not refer in the text of your manuscript. Please ensure that you refer to Table 5 in your text; if accepted, production will need this reference to link the reader to the Table.

Answer:

First of all, I am very grateful to the reviewers for their valuable comments, I will make the following reply：

In the revision of this paper, this problem has been corrected. 

7.Please upload a copy of Supporting Information Figures 1-4 and Supporting information Tables 1-5 which you refer to in your text on page 24.

Answer:

First of all, I am very grateful to the reviewers for their valuable comments, I will make the following reply：

In this revision, I will upload a copy of Supporting Information Figures 1-4 and Supporting information Tables 1-4.

Reviewer #1: 

2.Although the paper pays attention only to the changes and distribution characteristics of cultural heritage, it is still necessary to explain and discuss some data in detail: why did the number of cultural heritages increase sharply from the Song and Yuan dynasties?

Answer:

First of all, I am very grateful to the reviewers for their valuable comments, I will make the following reply：

The feudal state in the Song and Yuan Dynasties went from being divided to being united gradually, during this period, the economic center of gravity was moved to the south, the commodity economy had new development, the exchanges between China and foreign countries were frequent, overseas trade was developed, and science and technology, literature and art, and private trade were unprecedentedly prosperous, resulting in a substantial increase in the number of cultural heritage. 

3.The selection basis of different periods needs to be further elaborated: The Ming and Qing Dynasties were more than 100 years longer than the Song and Yuan Dynasties, and this period has been much longer than the modern period.

Answer:

First of all, I am very grateful to the reviewers for their valuable comments, I will make the following reply：

In the division of historical periods in this article, it is not divided evenly by the length of time, but is classified and divided according to the characteristics of the development stages of ancient Chinese history. 

The pre-Qin period was the formation period of ancient Chinese civilization, and it was the budding stage of the historical development of Chinese civilization. Chinese history moved from cave dwellings to settled farming, from the Stone Age to the Iron Age, from savagery and barbarism to civilization and civilization. There was no unified feudal dynasty, so they were grouped together. 

The Qin and Han dynasties were the period of the initial development of my country's feudal society and the first appearance of the unification situation. During this period, the formation of the unification situation of the Qin eradication of the six countries and the succession of the Qin Dynasty system by Emperor Wu of the Han Dynasty achieved great unification, and the social productivity has been greatly improved, so it is divided into together. 

The Wei, Jin, Southern and Northern Dynasties were a period of my country's feudal state division and great national integration. During this period, the country was divided for a long time, with various regimes standing side by side, wars against each other constantly, and there were several partial unifications, but the situation of division was the main one. So they were grouped together. 

During the Sui, Tang and Five Dynasties, my country's feudal society reached its peak and then split. It was also an important development period for my country's unified multi-ethnic state. Various ethnic groups were further integrated, the central government strengthened its jurisdiction over border areas, and the unified multi-ethnic state was further developed.

The Song and Yuan Dynasties were the period when the feudal country was divided and separated, and the national regimes were established side by side to gradually realize the great unification. For a long time, the two Song Dynasties were in a situation of split and separate multi-ethnic regimes. The Southern Song Dynasty and the Yuan Dynasty coexisted for a long time, and the Yuan Dynasty unified China and ended this situation. The commodity economy developed new development, the exchanges between China and foreign countries were frequent, and the folk trade was unprecedentedly prosperous. Various cultures were also acquired development during this period. 

During the Ming and Qing Dynasties, my country's feudal system gradually declined and the unified multi-ethnic state was further consolidated. During this period, the autocratic monarchy was unprecedentedly strengthened and reached its peak, which seriously hindered social progress. Facing the unprecedented prosperity of the commodity economy, the rulers still The implementation of the policy of emphasizing agriculture and suppressing commerce and closing the country has made Chinese culture and technology lag behind the development trend of the world. So divide it up together. 

The modern period refers to the period after the Qing Dynasty. Due to the short history and the period of great change and turmoil in the whole country, it is mainly based on modern practical activities and has a clear purpose and urban orientation, so it is divided into together. 

4.The paper uses a lot of space to introduce the composition of cultural heritage in different dynasties, but this part is not reflected in the abstract: It is needed to add a diagram that shows different parts and also their linkages and relationships.

Answer:

First of all, I am very grateful to the reviewers for their valuable comments, I will make the following reply：

In this revision, the composition and spatial distribution characteristics of cultural heritage of different dynasties have been integrated into the table, so that the temporal and spatial characteristics of cultural heritage in each period can be intuitively understood also their linkages and relationships. 

5.Although the paper has proposed to conduct in-depth research and analysis on the influencing factors in future research, it is still suggested to add some discussion on the results in the paper, there is too little discussion about the results.

Answer:

First of all, I am very grateful to the reviewers for their valuable comments, I will make the following reply：

In this revision, the discussion section focuses on the influencing factors of cultural heritage in the Yellow River Basin during the historical period. It supports the temporal and spatial distribution characteristics of cultural heritage in various historical periods. 

Reviewer #2: 

1. The script is too long and often redundant. For example, paragraphs 3.1 to 3.7 can be easily

summarized in one or more tables, limiting the text to the most crucial comments. It is also very demanding in terms of illustrations (Figures 2, 3 and 4, in particular, would need a lot of space, probably pages, to be fully readable).

Answer:

First of all, I am very grateful to the reviewers for their valuable comments, I will make the following reply：

In this revision, parts 3.1 to 3.7 have been integrated into the table, so that we can intuitively understand the temporal and spatial characteristics of cultural heritage in each period. And the illustrations in the paper have been standardized. 

2.Overall, its english is quite poor. Many expressions are probably literal, or almost literal translations of chinese constructions which, once translated, remain obscure.In short, my impression is that - as far as points 1 and 2 are concerned - the text should be shortened by 50% and the english reviewed if not re-written by a mother language speaker.

Answer:

First of all, I am very grateful to the reviewers for their valuable comments, I will make the following reply：

In this revision, the comments mentioned by the reviewers have been revised one by one, and the inappropriate and unreasonable words mentioned in the article have been deleted and revised to make the article more smooth and complete. At the same time, the article is more concise and clear by means of table integration. 

3.The Authors should attenuate some nationalistic overtones of their text. In particular, although the fact the the Yellow River Basin was the cradle of the Chinese civilization is universally accepted, sentences of the same content are unnecessarily repeated in different parts of the article (see in particular paragraph 3).

Answer:

First of all, I am very grateful to the reviewers for their valuable comments, I will make the following reply：

in this revision, the wording of nationalistic overtones has been reduced, and the article has been adjusted to make the article more streamlined and complete. 

4.Similarly, I find disturbing some very idealistic assumptions - like the statement that "Cultural heritage (is) the crystallization of the development of human civilization" or the concept of a "Chinese spiritual civilization" (see the first page). Such expressions, I fear, have nothing to do with science.

Answer:

First of all, I am very grateful to the reviewers for their valuable comments, I will make the following reply：

Similar unreasonable statements in the article have been deleted in this revision. 

5. The Authors have an evolutionary, firmly stadial concept of history. Instead of using archaeological patterns for reconstructing history, the give history for granted (the discussion of every historical period stats with a summary interpretation of an evolutionary step) and proceed to discuss the locational variations of sites.

Moreover, one of their periods is not credible: what is the historical sense of a pre-Qin stage, which goes from Paleolithic (1.700.000 mya, see paragraph 3.1) to the Qin dynasty (late 3rd century BC)? And sites which are linked to Revolution events, can be really analyzed in the same conceptual and analytical frame of prehistoric or protohistoric sites?

Answer:

First of all, I am very grateful to the reviewers for their valuable comments, I will make the following reply：

In the division of historical periods in this article, it is not divided evenly by the length of time, but is classified and divided according to the characteristics of the development stages of ancient Chinese history. 

The pre-Qin period was the budding stage of the historical development of Chinese civilization. Chinese history moved from cave dwellings to settled farming, from the Stone Age to the Iron Age, and from savagery and barbarism to civilization and enlightenment. During this period, the cultural heritage of the Yellow River Basin was dominated by ancient tombs, which laid the foundation of Chinese civilization. There was no unified feudal dynasty, so they were grouped together. 

The ruins of revolutionary events in the pre-Qin period in this article are all ancient ruins of wars formed by events such as war, separatism and regime change that occurred under the historical conditions at that time. It is classified as ancient ruins in the first eight batches of national cultural relics protection units in the country. 

5.Even though the spatial analysis seems to be accurate and interesting, the evolution of settlement patterns in the study area is represented in terms of simple dimensional point patterns in a simple void space. No geomorphology, no remote sensing, no paleofluvial variations, no climatic changes, no changing patterns of soil exploitation, no demography. What about changing production systems? How did a type of economical practices of a phase affected those of the following period? What happened to wood coverings and soil formations? This is what, in my view, makes the article very idealistic and far from a materialist historical effort.

Answer:

First of all, I am very grateful to the reviewers for their valuable comments, I will make the following reply：

In the discussion part of this revision, the influencing factors of cultural heritage in the historical period of the Yellow River Basin, such as topography and climate, river hydrology, ancient river course changes, administrative system, social economy, etc., have been discussed. It supports the temporal and spatial distribution characteristics of cultural heritage in various historical periods.

---

## [Decision Letter · Decision Letter 1]

28 Apr 2022

PONE-D-21-38789R1The spatial and temporal differentiation characteristics of cultural heritage in the Yellow River BasinPLOS ONE

Dear Dr. Jiao,

Thank you for submitting your manuscript to PLOS ONE. After careful consideration, we feel that it has merit but does not fully meet PLOS ONE’s publication criteria as it currently stands. Therefore, we invite you to submit a revised version of the manuscript that addresses the points raised during the review process.

ACADEMIC EDITOR:

The acknowledgments have some obvious errors. The manuscript was written by the cooperation of the four authors. Why thank Li Wei？ Li Wei is one of the authors, authors should not thank author themselves.

The word "references" is typed wrongly. I suggest all the authors to check all the language carefully.

We look forward to receiving your revised manuscript.

Kind regards,

Jian Liu

Academic Editor

PLOS ONE

Journal Requirements:

Reviewers' comments:

Reviewer's Responses to Questions

**Comments to the Author**

1. If the authors have adequately addressed your comments raised in a previous round of review and you feel that this manuscript is now acceptable for publication, you may indicate that here to bypass the “Comments to the Author” section, enter your conflict of interest statement in the “Confidential to Editor” section, and submit your "Accept" recommendation.

Reviewer #1: All comments have been addressed

Reviewer #2: All comments have been addressed

2. Is the manuscript technically sound, and do the data support the conclusions?

Reviewer #1: Yes

Reviewer #2: Yes

3. Has the statistical analysis been performed appropriately and rigorously? 

Reviewer #1: Yes

Reviewer #2: Yes

4. Have the authors made all data underlying the findings in their manuscript fully available?

Reviewer #1: Yes

Reviewer #2: Yes

5. Is the manuscript presented in an intelligible fashion and written in standard English?

Reviewer #1: Yes

Reviewer #2: Yes

6. Review Comments to the Author

Reviewer #1: (No Response)

Reviewer #2: The acknowlegments should be less romantic and more sythetic.

The word "references" is typed wrongly

7. PLOS authors have the option to publish the peer review history of their article (what does this mean?). If published, this will include your full peer review and any attached files.

Reviewer #1: No

Reviewer #2: No

---

## [Author Response · Author response to Decision Letter 1]

3 May 2022

First of all, I am very grateful to the reviewers for their valuable comments, I will make the following reply：

I have rewritten and revised the acknowledgments section to make it more accurate and complete. At the same time, I also corrected the obvious errors such as "references", and proofread the full text language.

---

## [Editor Report · Decision Letter 2]

4 May 2022

PONE-D-21-38789R2The spatial and temporal differentiation characteristics of cultural heritage in the Yellow River BasinPLOS ONE

Dear Dr. Jiao,

Thank you for submitting your manuscript to PLOS ONE. After careful consideration, we feel that it has merit but does not fully meet PLOS ONE’s publication criteria as it currently stands. Therefore, we invite you to submit a revised version of the manuscript that addresses the points raised during the review process.

ACADEMIC EDITOR:

I am still very curious, why three authors " Jianping Jiao, Jianwu Qi, Yujia Ma" were listed in the "7.Acknowledgments".

Jianping Jiao is the Corresponding author in the cover letter. As the Corresponding author, are you writing the paper yourself? Why thank yourself and other authors in the"7.Acknowledgments".

Authors should not thank author themselves in the "7.Acknowledgments" in the published paper.

Please delete them from the authors or delete them from the "7.Acknowledgments".

We look forward to receiving your revised manuscript.

Kind regards,

Jian Liu

Academic Editor

PLOS ONE

Journal Requirements:

Additional Editor Comments (if provided):

I am still very curious, why three authors " Jianping Jiao, Jianwu Qi, Yujia Ma" were listed in the "7.Acknowledgments".

Jianping Jiao is the Corresponding author in the cover letter. As the Corresponding author, are you writing the paper yourself? Why thank yourself and other authors in the"7.Acknowledgments".

Authors should not thank author themselves in the "7.Acknowledgments" in the published paper.

Please delete them from the authors or delete them from the "7.Acknowledgments".
---

## [Author Response · Author response to Decision Letter 2]

5 May 2022

First of all, I am very grateful to the reviewers for their valuable comments, I will make the following reply：

In the revision of the acknowledgments section, the three authors Jianwu Qi , Jianping Jiao and Yujia Ma were deleted as required, and the acknowledgments section was also revised.

---

## [Editor Report · Decision Letter 3]

9 May 2022

PONE-D-21-38789R3The spatial and temporal differentiation characteristics of cultural heritage in the Yellow River BasinPLOS ONE

Dear Dr. Jiao,

Thank you for submitting your manuscript to PLOS ONE. After careful consideration, we feel that it has merit but does not fully meet PLOS ONE’s publication criteria as it currently stands. Therefore, we invite you to submit a revised version of the manuscript that addresses the points raised during the review process.

ACADEMIC EDITOR:Please delete the first paragraph of the acknowledgments. Jian Liu is the editor of the paper, He did not know any author of the paper. It is impossible for him to help or teach you in the research or paper writing.

We look forward to receiving your revised manuscript.

Kind regards,

Jian Liu

Academic Editor

PLOS ONE

Journal Requirements:

Additional Editor Comments (if provided):

Please delete the first paragraph of the acknowledgments. Jian Liu is the editor of the paper, He did not know any author of the paper. It is impossible for him to help or teach you in the research or paper writing.

The authors have made obvious, different or even ridiculous errors in the manuscript for three times. Please consult someone who have published scientific papers and has experience of publishing scientific paper before.
---

## [Author Response · Author response to Decision Letter 3]

10 May 2022

First of all, I am very grateful to the editor for their valuable comments, I will make the following reply：

Regarding the acknowledgments of the last submission, Dr. Jian Liu mentioned in the revision is my classmate in Capital Normal University, not the editor of your journal. In this revision, I has been removed from Acknowledgments as required by this revision.

---

## [Editor Report · Decision Letter 4]

11 May 2022

The spatial and temporal differentiation characteristics of cultural heritage in the Yellow River Basin

PONE-D-21-38789R4

Dear Dr. Jiao,

We’re pleased to inform you that your manuscript has been judged scientifically suitable for publication and will be formally accepted for publication once it meets all outstanding technical requirements.

Kind regards,

Jian Liu

Academic Editor

PLOS ONE

Additional Editor Comments (optional):

All the comments have been addressed.
---

## [Editor Report · Acceptance letter]

19 May 2022

PONE-D-21-38789R4 

The spatial and temporal differentiation characteristics of cultural heritage in the Yellow River Basin 

Dear Dr. Jiao:

I'm pleased to inform you that your manuscript has been deemed suitable for publication in PLOS ONE. Congratulations! Your manuscript is now with our production department. 

Kind regards, 

on behalf of

Dr. Jian Liu 

Academic Editor

PLOS ONE